# The Effects of Maternal Iron and Folate Supplementation on Pregnancy and Infant Outcomes in Africa: A Systematic Review

**DOI:** 10.3390/ijerph21070856

**Published:** 2024-06-29

**Authors:** Yibeltal Bekele, Claire Gallagher, Don Vicendese, Melissa Buultjens, Mehak Batra, Bircan Erbas

**Affiliations:** 1School of Psychology and Public Health, La Trobe University, Melbourne, VIC 3086, Australia; m.buultjens@latrobe.edu.au (M.B.); m.batra@latrobe.edu.au (M.B.); b.erbas@latrobe.edu.au (B.E.); 2School of Public Health, Bahir Dar University, Bahir Dar 79, Ethiopia; 3School of Population and Global Health, The University of Melbourne, Melbourne, VIC 3010, Australia; claire-gallagher@live.com (C.G.); don.vicendese@unimelb.edu.au (D.V.); 4School of Computing, Engineering and Mathematical Sciences, La Trobe University, Melbourne, VIC 3086, Australia

**Keywords:** adverse birth outcomes, neonatal mortality, infant mortality, iron, folate, Africa

## Abstract

Background: Iron and folate deficiency are prevalent in pregnant women in Africa. However, limited research exists on the differential effect of oral iron-only, folate-only, or Iron Folic Acid (IFA) supplementation on adverse pregnancy and infant outcomes. This systematic review addresses this gap, focusing on studies conducted in Africa with limited healthcare access. Understanding these differential effects could lead to more targeted and potentially cost-effective interventions to improve maternal and child health in these settings. Methods: A systematic review was conducted following PRISMA guidelines. The primary exposures were oral iron-only, folate-only, or IFA oral supplementation during pregnancy, while the outcomes were adverse pregnancy and infant outcomes. A qualitative synthesis guided by methods without meta-analysis was performed. Results: Our qualitative synthesis analysed 10 articles reporting adverse pregnancy (adverse birth outcomes, stillbirths, and perinatal mortality) and infant outcomes (neonatal mortality). Consistently, iron-only supplementation demonstrated a reduction in perinatal death. However, evidence is insufficient to assess the relationship between iron-only and IFA supplementation with adverse birth outcomes, stillbirths, and neonatal mortality. Conclusion: Findings suggested that iron-only supplementation during pregnancy may reduce perinatal mortality in African women. However, evidence remains limited regarding the effectiveness of both iron-only and IFA supplementation in reducing stillbirths, and neonatal mortality. Moreover, additional primary studies are necessary to comprehend the effects of iron-only, folate-only, and IFA supplementation on pregnancy outcomes and infant health in the African region, considering rurality and income level as effect modifiers.

## 1. Introduction

Worldwide, an estimated 2.5 million newborn [1] and 3.6 million infant [2] deaths occur each year, collectively representing 47% and 75% of all reported deaths among children under the age of five, respectively [3]. These losses are predominantly concentrated in low- and middle-income countries (LMIC), with 41% and 55% of global neonatal and infant deaths occurring in the Sub-Saharan Africa region alone [1,3]. Encouragingly, the past three decades have witnessed a significant decline in the global neonatal and under-five death rates, with neonatal mortality reducing by 52% between 1990 and 2017 (from 36.6 deaths to 18 deaths per 1000 live births) and under-five mortality reducing by 59% between 1990 and 2019 (from 93 deaths to 57 deaths per 1000 live births). However, the reduction in neonatal mortality has been much slower in Africa, particularly in the Sub-Saharan region, decreasing from 45.3 per 1000 live births in 1990 to 27.5 per 1000 live births in 2019 [1,3]. Furthermore, it is predicted based on current trajectories that approximately 27.8 million neonatal deaths will occur between 2018 and 2030, with the majority anticipated in South Asia and Africa [1].

In Africa, adverse birth outcomes, neonatal mortality, and infant mortality remain a main challenge, with variations across regions. For example, in Nigeria, the neonatal mortality rate was stagnant at 41 per 1000 live births from 1990 to 2013, with rural areas facing higher rates (42 per 1000 live births) than urban areas (31 per 1000 live births) [4]. Similarly, in Ghana, rural areas had significantly higher neonatal deaths at 63.6 per 1000 live births compared to 36.4 per 1000 live births in urban areas [5]. Another study in Nigeria found infant mortality rates of 70 per 1000 live births in rural areas and 49 per 1000 live births in urban areas [6]. This persistent challenge hinders the region’s progress toward national and international goals, particularly in rural areas where access to healthcare and enablers of health literacy are much more challenging [7,8].

According to a study on Sustainable Development Goals (SDGs) progress, it is projected that 71% (145 out of 204) and 80% (164 out of 204) of countries will successfully meet the SDGs aimed at reducing neonatal and under-five mortality by 2030, respectively [9]. However, 30% (61 out of 204) of countries are not on track to meet these goals, with most of them (43 out of 61) located in the Sub-Saharan African region [9]. Challenges such as poor maternal healthcare service utilisation [10,11], financial hardships [11], health illiteracy [12], weak healthcare systems [12], limited access to and availability of services [12,13], and poor quality of maternal healthcare services [14] are major obstacles hindering the achievement of SDG in the Sub-Saharan region. Evidence suggests that early and regular utilisation of existing maternal health care services, including perinatal care, is crucial for reducing health challenges faced by mothers and newborns, especially in the Sub-Saharan regions [15].

Iron plays a crucial role in numerous metabolic functions critical for pregnancy, including oxygen transportation, DNA synthesis, and electron transport [16]. The demand for iron significantly increases during pregnancy to support both fetal growth and development and maternal physiological adaptations [17]. Similarly, folate is essential for cell division and growth and crucial for both fetal development and placental function [18]. While studies have established associations between iron and/or folate deficiencies and pregnancy outcomes [19,20], the specific mechanisms by which supplementation with iron or folate, alone or combined, influence these outcomes remain unclear. This systematic review aims to synthesise existing evidence on the differential effects of iron-only, folate-only, and IFA supplementation to elucidate potential causal pathways linking these interventions to improved pregnancy outcomes in Africa.

In Africa, particularly in the Sub-Saharan Africa region, nutritional issues such as food insecurity and malnutrition, including iron deficiency, are prevalent [21,22,23]. Adequate intake of iron and folic acid is crucial for healthy development, disease prevention, and overall well-being [24,25]. A study in Tanzania, Africa, found an increased risk of stillbirth and perinatal mortality associated with iron deficiency anaemia [26]. Likewise, studies from other low-income countries and southern Asia show that iron and folic acid supplementation reduces the odds of adverse infant outcomes, including neonatal mortality [27,28] and infant mortality [29]. To mitigate the gap, the World Health Organization (WHO) recommends daily supplementation of oral iron and folic acid with 30 to 60 mg of element iron and 0.4 mg of folic acid for all pregnant women throughout their pregnancy to prevent maternal anaemia, sepsis, low birth weight, and preterm birth [30]. Furthermore, global evidence shows iron and/or folate supplementation can reduce the risk of miscarriages, stillbirths, perinatal mortality, neonatal mortality, and infant mortality [31,32,33,34,35].

The coverage of iron and folate supplementation improved over time in Africa. Data from national demographic health surveys indicated that coverage of iron supplementation reached 60% in Ethiopia [36], 69% in Nigeria [37], and 86% in Uganda [38]. Additionally, studies, including systematic reviews and meta-analyses conducted in Africa, identified adherence as a major issue [39,40]. For example, a population-based study conducted in Sub-Saharan Africa showed that non-adherence to IFA supplementation in the region ranges from 27% in Senegal to 98.6% in Burundi [40]. Similarly, a systematic review conducted in Sub-Saharan Africa showed that compliance with iron and folate supplementation ranges from 10.6% in Kenya to 79% in Mozambique [39]. A study conducted in Niger among 923 pregnant women showed that 43.6% received IFA supplementation, of which 68.8% adhered to the recommended IFA supplementation [41]. Similarly, a study conducted in Ethiopia showed that the coverage of IFA was 60%, ranging from 18% in the Somali region to 83.3% in the Tigray region, whereas only 20% of them adhered to IFA supplementation [42]. Furthermore, there was a huge gap in adherence to IFA supplementation between rural areas (18%) and urban areas (66%) in Ethiopia [43,44].

Previous systematic reviews and meta-analyses have been conducted globally in 2000, 2012, and 2013, incorporating studies from randomised controlled trials assessing intervention impacts [45,46,47]. The findings of these reviews were limited, indicating that neither iron-only nor IFA supplementations were significantly associated with reductions in perinatal and neonatal deaths. It is worth noting that while some African countries may have been included in these reviews, none of the studies [48,49,50] reported perinatal death, neonatal death, or infant death as outcomes. As a result, the conclusions drawn from these reviews were primarily based on studies conducted outside of the African region, limiting their applicability to the unique context and challenges faced by African populations.

As far as we are aware, there has been no systematic review conducted in the African region regarding adverse pregnancy and infant outcomes and the role of these two key supplements during pregnancy. Therefore, in this review, we aimed to qualitatively synthesise any effects of iron-only, folate-only, and IFA oral supplementation during pregnancy on adverse pregnancy outcomes (such as adverse birth outcomes, stillbirths, small for gestational age, and perinatal mortality) and infant outcomes (neonatal mortality and infant mortality) in the African region. The outcome of this review is intended to provide valuable insight for programmers, policymakers, and researchers involved in addressing maternal and child health problems in the region.

## 2. Methods

This systematic review was conducted following the preferred Reporting Items for Systematic Review and Meta-Analysis (PRISMA-P) 2020 guidelines [51] (Appendix A). This review has been registered in the PROSPERO International Prospective Register of Systematic Reviews under the registration number CRD42023452588.

### 2.1. Selection Criteria

#### Types of Studies, Context, and Language

All quantitative studies, including randomised clinical trials, non-randomised clinical trials, and observational study designs such as cohort, case-control, and cross-sectional studies conducted in the African region were included. This review only includes full articles conducted in English, whereas study protocols, editorial reports, case reports, case series, conference papers, and abstracts without full articles were excluded.

### 2.2. Population of Interest

Our population of interest was healthy pregnant women residing in Africa. Women of any age, parity, or gestational stage were eligible for inclusion.

### 2.3. Exposure/Intervention and Comparator

#### 2.3.1. Exposure or Intervention

This review included studies that investigated the effects of oral iron-only, folate-only, or IFA supplementation during pregnancy. Studies involving fortification, combination with other micronutrients, and measured serum iron or folate levels as the intervention or exposure were excluded from the review as we were primarily interested in oral exposure’s effects, given its universal comparison across different countries.

#### 2.3.2. Comparators

The comparator group consisted of women who either took a placebo or did not receive any micronutrient supplementation.

### 2.4. Outcomes

#### 2.4.1. Adverse Pregnancy Outcomes

Adverse birth outcomes encompass a range of complications that can occur during pregnancy or childbirth. These include stillbirth (pregnancy termination after 28 weeks), small for gestational age (birth weight below the 10th percentile for gestational age), preterm birth (delivery before 37 weeks), low birth weight (birth weight less than 2500 g), perinatal mortality (fetal death after 28 weeks or newborn death within the first week), and neonatal mortality (newborn death within the first 28 days) [52,53]. Notably, adverse birth outcomes are assessed using a combination of more than two of these indicators.

#### 2.4.2. Infant Outcomes

Neonatal death is defined as mortality within the first 28 days, while infant mortality occurs within the first year [52,53].

### 2.5. Information Sources and Search Strategy

The relevant studies were retrieved from the following electronic databases on 29 August 2023: MEDLINE, PsycINFO, Embase, Scopus, CINAHL, Web of Science, and Cochrane. No time restrictions were applied during the search in these databases. Additionally, Google Scholar and Google Advanced Search were utilised to access grey literature and additional literature. The final search term used for the MEDLINE database to retrieve relevant articles is attached as a Appendix A. The last Google Scholar search was conducted on 07/03/2024 to ensure that the studies included were as recent as possible.

### 2.6. Study Selection

All collected articles from databases were exported to Covidence, a web-based software that assists researchers in conducting title, abstract, and full-text screening [54]. After importing to Covidence, all duplicated articles were removed. Authors (YB and CG) independently screened all titles and abstracts, and articles deemed relevant underwent full-text review. Any disagreement regarding the inclusion and exclusion of the articles was resolved by discussion or consultation with the senior author, BE.

### 2.7. Data Extraction and Quality Appraisal

YB and CG separately conducted data extraction and quality appraisal. Data extraction included the author’s first name, publication year, location, study setting, study design, population characteristics, rural population, sample size, types of exposure, dosage of exposure, duration of exposure, types of outcomes, confounder variables, effect estimate, and 95% confidence interval from each included study. Cochrane Effective Practice and Organisational Care (EPOC) [55] and the Newcastle-Ottawa Scale (NOS) risk of bias [56] assessment tools were used to assess the quality of the included studies for clinical trials and observational study design, respectively. Overall quality was categorised as “high risk of bias” if the study scored 0 to 3, “medium risk of bias” if the study scored 4 to 6, and “low risk of bias” if the study scored 7 to 9. Any discrepancies in rating between authors were resolved by discussion or with the agreement of the senior author, BE.

### 2.8. Data Synthesis and Meta-Analysis

A meta-analysis was not conducted due to the insufficient number of articles meeting the minimum criteria for each outcome. Instead, qualitative synthesis was performed following the Synthesis Without Meta-analysis (SWiM) reporting guidelines [57]. All articles were grouped based on their reported outcomes (adverse birth outcomes, perinatal, neonatal mortality, and infant mortality), with studies that reported multiple outcomes grouped into each corresponding outcome category. Within each group, the characteristics and effect estimate of each study are summarised and reported.

## 3. Result

### 3.1. Seach Results and Characteristics of Included Studies

A total of 3349 studies were initially retrieved from seven databases, and two studies were included from Google Advanced search. Among them, 1175 studies were excluded due to duplication, and 2095 studies were excluded during the title and abstract screening. Seventy-one studies were excluded during the full-text review because they lacked relevant outcomes, exposures, comparators, or study designs, or because they consisted only of abstracts without full articles (Figure 1). A total of 10 studies were eligible for inclusion. Of these, four were conducted in Ethiopia [33,58,59,60], two in Sudan [61,62], one in Cote d’Ivoire [63], and three involved multiple countries in the Sub-Saharan Africa region (Zimbabwe, Malawi, Cameroon, Uganda, Burkina Faso, Madagascar, Tanzania, Liberia, Benin, Congo, Niger, Zambia, Senegal, Chad, Guinea, the Democratic Republic of the Congo, Ghana, Mali, Nigeria, Angola, Burundi, Gambia, Kenya, Mauritania, and Sierra Leone) and the West African region (Ivory Coast, Mali, Senegal, Niger, Mauritania, and Burkina Fano) [64,65,66]. Regarding study design, four were cross-sectional [60,61,64,65], two were case-control [33,59], and four were cohort studies [58,62,63] (Table 1). The analysis included a total of n = 218,232 women who gave birth. Urban–rural status was not reported in two studies conducted among 6 West African countries and 19 Sub-Saharan countries [58,64], but among those that did, approximately 38% (n = 82,903) resided in rural areas. Even though studies did not include urban–rural definitions, the criteria commonly employed in the region include lifestyle, population density, economic activities, infrastructures, settlement patterns, industrialisation, and correlation with poverty [67]. Six studies collected information from the community [61,62,63,64,65,66], three studies from healthcare facilities [33,59,60], and one study utilised both (individual characteristics were collected from house-to-house interviews while health service utilisation, pregnancy outcomes, and neonatal health conditions were collected by reviewing health facility registration books) [58] (Table 1).

### 3.2. Interventions/Exposures

Overall, six studies assessed the effects of oral IFA supplementation [58,59,60,63,64,65], whereas four studies assessed the effects of iron-only supplementation [33,61,62,66]. Notably, none of the studies assessed folate-only supplementation. Furthermore, none of the included studies assessed the dose–effect response or the duration of supplementation (Table 2).

### 3.3. Quality Assessment

The quality assessment results showed that eight studies were at low risk of bias [33,58,60,61,63,64,65,66] and two studies were at medium risk of bias [59,62] (Table 1). The reasons for categorising medium risk of bias were related to participant selection, ascertainment of exposures, and outcomes. For instance, Hailemichael et al. [59] selected cases and control groups from healthcare facilities without providing detailed information on the definitions used to confirm preterm births and stillbirths. Similarly, in the study by Ibrahim et al. [62], the method section lacked clarity, particularly concerning the tools utilised to collect exposure. One study did not include all potential confounders (such as history of antenatal care, maternal age, residence, and maternal educational status) in the final model [62]. Six studies collected information regarding perinatal oral supplementation through maternal recall after birth, which may increase the risk of recall bias [33,59,60,61,64,65] (Appendix A). Furthermore, the internal validity of the study by Kone et al. [63] might have been impacted by social desirability bias, leading to misclassification, as some stillbirths might have been classified as early neonatal deaths due to challenges in distinguishing perinatal asphyxia, influenced by common cultural interpretations in Sub-Saharan Africa.

### 3.4. Outcomes (n = 15)

Among the included studies, there were nine reported pregnancy outcomes, including “adverse birth outcome” (n = 3) [58,59,63], stillbirths (n = 5) [59,60,62,63,66], and perinatal deaths (n = 4) [33,61,62,63]. The other three studies considered infant outcomes, specifically neonatal death (n = 3) [62,64,65] (Table 2). However, no studies reported small for gestational age or infant death as an outcome.

#### 3.4.1. Adverse Pregnancy Outcomes (n = 12)

##### Adverse Birth Outcomes (n = 3)

Two of the three studies found positive associations between oral IFA supplementation and reduced adverse birth outcomes. One study reported an adjusted odds ratio (aOR) of 0.52 (95% CI; 0.41 to 0.68) [58]. The other study found an aOR of 3.15 (95% CI; 1.71 to 5.80) [63], indicating a possible protective effect of IFA supplementation. A low-risk study conducted by Zelka et al. [58] among 2402 pregnant women in Ethiopia reported that 66.11% of the participants attended antenatal care services, and 76.29% received IFA supplementation. Similarly, a study conducted by Kone et al. among 2976 pregnant women in Cote d’Ivoire indicated that 87.2% of the participants resided in rural areas, 96.3% of participants received antenatal care, and 72.0% received IFA supplementation. However, the association of iron-only or IFA supplementation with adverse birth outcomes was not significant in the other study [59]. A medium-risk study conducted by Hailemichael et al. [59] in Ethiopia among 405 mothers who gave birth revealed that 26.4% resided in rural areas, 58% received four or more antenatal care visits, and 19% did not take IFA supplementation [59]. The finding showed that IFA supplementation was not statistically associated with the odds of reducing adverse birth outcomes (aOR 0.67; 95% CI; 0.18 to 2.57) [59]. The potential reasons for these non-significant results in this case-control study might be attributed to factors such as recall bias, social desirability bias, or low adherence rates reported among the participants (29.23%) [59] (Table 2).

It is important to note that there were variations in the case definitions of adverse birth outcomes across each of the studies. For instance, Zelka et al. [58] measured adverse birth outcomes using abortions, low birth weight, or preterm birth, while Kone et al. [63] used a combination of the following criteria: abortion, stillbirths, or perinatal mortality. On the other hand, Hailemichael et al. [59] measured adverse birth outcomes using low birth weight, preterm birth, and stillbirths (Table 2).

##### Stillbirths (n = 5)

Five studies assessed the relationship between oral iron-only or IFA supplementation on stillbirths [59,60,61,63,66]. A low-risk study by Lolaso et al. [60] in Ethiopia reported a significant protective effect of IFA supplementation. This study included a sample of 1980 pregnant women, of whom 45.7% resided in rural areas, 92.2% received antenatal care services, and 84% received IFA. The finding showed that the odds of stillbirth were significantly greater in those who did not take IFA supplementation compared with those who did (aOR 8.26; 95%CI; 4.82 to 14.16). In contrast, a prospective cohort study by Chalumeau et al. [66] among 20,326 mothers in six West African countries found that 28.4% received antenatal care services, and 12.4% received iron-only supplementation. The absence of iron-only supplementation showed a slight, non-significant increase in stillbirth odds (cOR 1.20; 95% CI; 0.90 to 1.70) [66]. Possible reasons for this lack of significance at *p* = 0.05 might include unaccounted confounding factors, self-reported iron supplementation, and a 5.8% loss to follow-up. Another cohort study conducted by Ibrahim et al. [61], a case-control study by Hailemichael et al. [59], and a cohort study conducted by Kone et al. [63] indicated that iron-only supplementation and IFA were not associated with adverse birth outcomes, including stillbirths (Table 2). However, stillbirths were not assessed separately in the latter studies; instead, they were merged with other adverse outcomes, so a definitive conclusion cannot be drawn.

##### Perinatal Death (n = 4)

Studies conducted by Goba et al. [33] and Ali et al. [61] estimated the association between iron-only supplementation and perinatal death, with both reporting significant protective effects with aOR 3.30 (95% CI; 1.16 to 9.76) and aOR 0.06 (95% CI; 0.02 to 0.10), respectively. Goba et al.’s [33] study in Ethiopia included 378 women who gave birth, revealing that 56.1% resided in rural areas, 92.6% received antenatal care, and 80.2% received iron-only supplementation. Ali et al.’s study [61] in Sudan among 808 pregnant women showed that 60.8% of participants resided in urban areas and 41.8% attended at least one antenatal care visit (Table 2). However, studies conducted separately by Ibrahim et al. [62] and Kone et al. [63] reported that iron-only and IFA supplementation, respectively, were not associated with adverse birth outcomes that included perinatal mortality, but merging outcomes limits further interpretation.

#### 3.4.2. Infant Outcomes (n = 3)

##### Neonatal Death (n = 3)

Two out of three studies [64,65] conducted in Sub-Saharan countries employed data from large, nationally representative Demographic and Health Surveys (DHS) to examine the relationship between oral IFA supplementation and neonatal mortality. Titaley et al. [65] investigated 100,683 mothers living in Sub-Saharan countries, showing that 72% resided in rural areas, 23.1% did not receive antenatal care, and 33.1% did not take IFA supplementation. Their findings indicated that IFA supplementation was associated with a reduced risk of neonatal death (aHR 0.81; 95% CI; 0.71 to 0.94) among all populations [65]. Godha et al. [64] examined 82,203 mothers, revealing varied rates of IFA supplementation uptake during pregnancy, ranging from 3% in Gambia to 57% in Burundi for non-uptake and from 1% in Burundi to 61% in Zambia for uptake lasting 90 or more days. They excluded an initial observed bivariate negative association between IFA supplementation and neonatal death (the effect size was not reported) from the final multivariate model, raising concerns about potential bias [64]. Additionally, a study by Titaley et al. included DHS (Demographic Health Survey) data from 2003 to 2007, whereas Godha et al. included DHS data from 2014 and later [64] (Table 2). However, neither of the studies conducted stratification analysis by urban or rural residence, educational status, or income level. Similarly, in another study conducted by Ibrahim et al. [62], after controlling age, educational status, and parity, iron-only supplementation was not found to be statistically associated with adverse birth outcomes that included neonatal death (the effect size was not reported) (Table 2). However, this study did not provide additional information about any other characteristics.

## 4. Discussion

This systematic review comprehensively synthesises evidence incorporating randomised trials and observational studies on the effects of iron-only and IFA oral supplementation during pregnancy on adverse pregnancy and infant outcomes in Africa, irrespective of dose and duration of supplementation. Although the data did not permit us to conduct a meta-analysis, the findings of the review indicated a protective effect of iron-only supplementation in reducing perinatal mortality. However, the findings were insufficient to assess the relationship between iron-only and IFA supplementation in reducing the odds of stillbirths, adverse birth outcomes, and neonatal mortality.

This review identified two studies [58,63] that consistently reported that IFA supplementation was positively associated with reducing the likelihood of adverse birth outcomes, while a study by Hailemichael et al. [59] reported that IFA supplementation showed no association with the reduction in adverse birth outcomes. The possible reason for this variation might be attributed to the differences in the definitions of the outcomes considered (adverse birth outcomes). For instance, a study conducted by Kone et al. [63] defined adverse birth outcomes as any one of abortion, stillbirths, or perinatal mortality, while Hailemichael et al. [59] used low birth weight, preterm birth, and stillbirths as criteria for defining adverse birth outcomes. Furthermore, variations in the study design among the included studies may also contribute to the difference. For example, studies conducted by Zelka et al. [58] and Kone et al. [63] used a cohort design, while Hailemichael et al. [59] used a case-control design, making them more likely to be impacted by recall bias. None of the included studies performed a stratification analysis by variables representing social determinants of health [58,59,63]. This is important because factors like urban/rural residence, educational status, age, and income can potentially influence the effects of oral IFA supplementation during pregnancy on birth outcomes.

Two studies inconsistently reported the relationship between iron-only and IFA supplementation in stillbirths. One study conducted by Lolaso et al. in Ethiopia [60] reported that iron-only supplementation reduced the odds of stillbirths. However, a study conducted by Chalumeau et al. in West Africa [66] reported that iron-only supplementation was not statistically associated with stillbirths. These discrepancies could be attributed to variations in the characteristics of the study participants and confounding adjustments. For example, a study conducted by Lolaso et al. [60] among 1980 women revealed that 16% did not receive IFA supplementation, while in a study by Chalumeau et al. [66], only 11.4% of participants received iron-only supplementation during pregnancy. Additionally, Chalumeau et al. [66] provided a crude estimate without adjusting for potential confounders, which could result in a biased estimate.

Another outcome identified distinctly within the studies included in this review is perinatal death. The findings of the review identified two studies that found iron-only supplementation was associated with reduced odds of perinatal death [33,61]. Even though the directions of association were similar between studies conducted by Goba et al. and Ali et al. [33,61], there was substantial variation in the odds ratios, which were 0.06 [61] and 0.30 [33]. This may be due to the differences in sample size, study design, and characteristics of the study participants. For instance, Goba et al. [33] included 378 women who gave birth, while Ali et al. included 808 participants [61]. Additionally, a cross-sectional study conducted by Ali et al. [61] reported that 60.8% of the participants resided in urban areas, whereas in a case-control study conducted by Goba et al., nearly 56% of the participants resided in rural areas [33]. This urban–rural participant difference between the included studies may lead to differences in service accessibility and health literacy [33,61]. The limited data available in this review indicate a potential association between iron-only supplementation and reduced perinatal mortality in African women. However, further research is necessary to confirm these findings and elucidate the underlying mechanism.

We identified three studies [62,64,65] examining the effects of oral iron-only and IFA supplementation during pregnancy on neonatal death, yielding inconsistent results. This discrepancy is attributed to the differences in the characteristics of study participants and selection criteria. The study by Titaley et al. [65] used criteria of 80% population-level malaria prevalence to define malaria-endemic countries, while the study by Godha et al. [64] used a criterion based on countries implementing antimalaria prophylaxis during pregnancy. It is important to note that only 66% of malaria-endemic countries in Africa implement antimalaria prophylaxis treatment during pregnancy, and just 25% of pregnant women in the region have access to antimalaria prophylaxis during pregnancy [68]. This selection criterion has the potential to exclude malaria-endemic countries that do not implement antimalaria prophylaxis treatment. Moreover, the study by Titaley reported that 72% of the participants resided in rural areas, nearly 51% were illiterate, 77.4% received antenatal care services, and 65.9% took iron folate supplementation [65]. However, the study by Godha et al. did not report any descriptive characteristics of the participants [64].

In general, there is a lack of studies conducted in Africa, and the study design employed often falls within the lower level of the evidence hierarchy. Additionally, no evidence was found regarding the impact of iron-only, folate-only, and IFA supplementation on small for gestational age and infant mortality. This lack of evidence may impede the data generation decisive for informing policy decisions. Interestingly, the absence of evidence on the impact of folic acid supplementation itself may indicate folic acid in this region being primarily available through fortified food sources, warranting further investigation.

Some African countries offer free iron and/or folate supplementation programs [69]. However, program effectiveness is hampered by limited access. This includes the insufficient supply of supplements, inadequate counselling on their benefits and appropriate use, and challenges for pregnant women to adhere to supplementation schedules [70]. Furthermore, structural barriers also hinder uptake. Lack of leadership engagement, weak family support systems, and limited access to antenatal care services restrict program reach. Additionally, misconceptions about supplementation, fear of side effects, and social norms that discourage its use create further hurdles that should be addressed as there is a clear need for improvements in health literacy [71,72].

This systematic review is subject to several limitations. The first limitation of this review is the inclusion of studies published only in the English language; it may lead to the exclusion of potential articles published in languages other than English in Africa. The other limitation of this review is that all the included studies are observational study designs, which may limit the ability to draw a clear causal relationship between iron-only and IFA supplementation on adverse birth outcomes, stillbirths, perinatal death, and neonatal death. Additionally, none of the studies conducted stratification analysis based on urban/rural status, age, educational status, or income status. Furthermore, the studies did not include information on how long the supplementation was taken, the dosage, or the stage of pregnancy at which they started to take supplementation. This lack of information may hinder the comprehension of the effects of each supplementation on adverse birth outcomes, perinatal death, and neonatal death. Moreover, certain articles aggregated and reported different outcomes together as “adverse birth outcomes”, thereby limiting direct comparisons and concluding similar outcomes.

## 5. Conclusions and Recommendation

This review found that even oral iron-only supplementation during pregnancy has a positive impact on reducing perinatal death. However, the existing evidence is insufficient to establish a clear relationship between iron-only or IFA supplementation on adverse birth outcomes, stillbirths, and neonatal death. Additionally, no evidence was found to suggest the relationship between iron-only, folate-only, and IFA supplementation and outcomes such as small for gestational age and infant mortality. The limited studies we reviewed suggest that direct, cost-effective strategies can potentially mitigate adverse effects for pregnant women on a population scale. However, it is crucial to consider the potential limitations of such approaches due to the population paradox. Our focus on existing research may not fully capture the impact of individual variations in iron and anaemic status.

## Figures and Tables

**Figure 1 ijerph-21-00856-f001:**
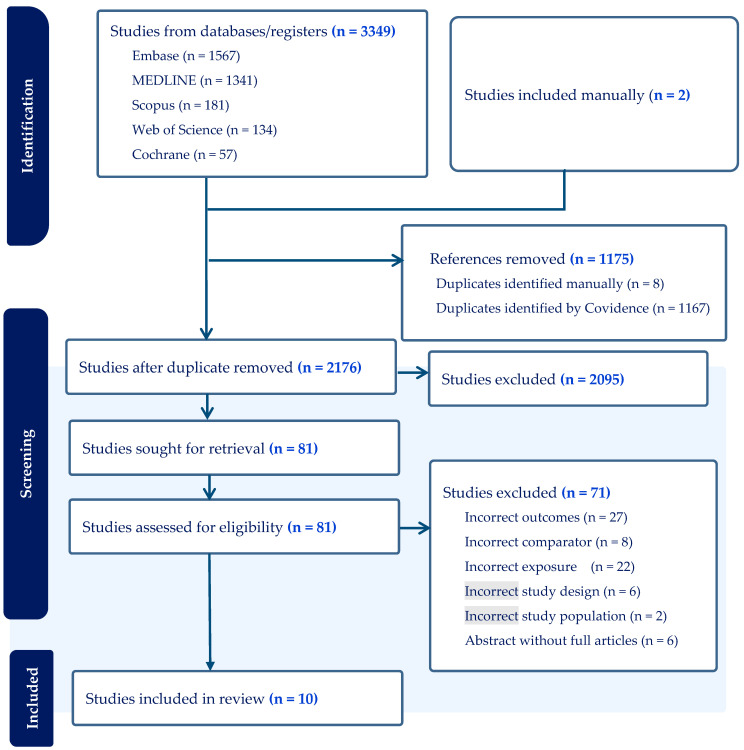
PRISMA (Preferred Reporting Items for Systematic Review and Meta-Analysis) flow chart diagram of studies searched for systematic review and meta-analysis in Africa, 2023. Footnote: “Incorrect” refers to not meeting inclusion criteria for a systematic review.

**Table 1 ijerph-21-00856-t001:** Characteristics of included articles conducted in the Africa region from 1994 to 2023.

First Author, Pub. Year (Ref)	Country (Setting) Year Study	Design/(Case to Control Ratio)	Study Population	Objective of the Study	Sample Size		Outcome Assessment	Quality Assessment Score
					Total Sample Size	Rural Participant		
Hailemichael 2020 [59]	Ethiopia (Facility-based) 2015	Unmatched Case-control (1 to 2)	Mothers who gave birth	To identify the determinants of adverse birth outcomes.	405	107 (26.4%)	Adverse birth outcomes measured by preterm birth (delivery before 37 completed weeks), low birth weight (less than 2.5 kg), and stillbirth (loss of foetus after 28 weeks of gestation).	Medium risk
Kone 2018 [63]	Côte d’Ivoire (Community-based) 2011 to 2014	Cohort	Pregnant women	To investigate pregnancy-related morbidities and factors associated with fatal outcomes.	2976	2594 (87.2%)	The foetal outcome was measured by abortion (termination before 28 weeks of gestation), stillbirth (foetal death after 28 weeks of gestation), and prenatal mortality (death after 28 weeks of gestation and before one week of the neonatal period).	Low risk
Ibrahim 1994 [62]	Sudan (Community-based) 1985 to 1988	Prospective cohort	Pregnant women	To identify factors associated with high-risk perinatal and neonatal mortality.	6275	6275 (100%)	Stillbirths (foetal death after 28 weeks of gestation), perinatal death (foetal death after 28 weeks of gestation or neonatal death within the 1st week), and neonatal death (death of the newborn within the 1st month of age).	Medium risk
Titaley 2010 [65]	Sub-Saharan Africa (Community-based) 2003 to 2007	Cross-sectional	Singleton live-born infants	To examine the association between iron/folic acid supplementation and perinatal antimalaria prophylaxis on neonatal mortality.	100,683	72,492 (72%)	Neonatal death was measured as the death of a neonate before 1st month of age.	Low risk
Godha 2022 [64]	Sub-Saharan Africa (Community-based) 2014	Cross-sectional	Mothers who had younger children under 2 years	To assess the association of antenatal IFA supplementation and malaria prophylaxis with neonatal mortality in Sub-Saharan Africa.	82,203	Urban–rural breakdown not reported	Neonatal mortality was measured by whether a participant experienced the loss of a newborn before 1 month of age.	Low risk
Goba 2017 [33]	Ethiopia (Facility-based) 2016	Unmatched Case-control (1 to 2)	Women who gave birth	To identify perinatal mortality risk factors.	378	213 (56.1%)	Perinatal mortality was measured by stillbirth (loss after 28 weeks of gestation) or early neonatal mortality (loss of the neonate within the first week of gestation).	Low risk
Ali 2014 [61]	Sudan (Community-based) 2010 to 2011	Cross-sectional	Pregnant women	To investigate factors associated with perinatal mortality.	808	317 (39.2%)	Perinatal death was defined as pregnancy loss occurring after 7 completed months of gestation (stillbirth) or death within the first week of delivery of a live-born neonate (early neonatal death).	Low risk
Zelka 2023 [58]	Ethiopia (Community and facility-linked) 2020 to 2021	Prospective Cohort	Pregnant women	To determine the effectiveness of the continuity of ANC and the determinats of adverse birth outcomes.	2198	Urban–rural breakdown not reported	Adverse pregnancy outcomes were measured by abortion (termination before 28 week or less than 1000 g), low birth weight (birth weight less than 2.5 kg), or preterm birth (delivery before 37 completed weeks) during delivery.	Low risk
Chalumeau 2002 [66]	West Africa (Community-based) 1994 to 1996	Prospective Cohort	Pregnant women	To identify late antenatal and intranatal risk factors for perinatal mortality and stillbirths.	20,326	Urban–rural breakdown not reported	Stillbirth was defined as a product of conception weighing greater than or equal to 500 g or with a gestational age > 22 weeks without evidence of life at birth.	Low risk
Lolaso 2021 [60]	Ethiopia (Facility-based) 2018	Cross-sectional study	Delivered women	To assess the magnitude and associated factors of stillbirth.	1980	905 (45.7%)	Stillbirth was defined as a baby born dead at 28 weeks of gestation or more, with a birth weight of ≥1000 g or a baby length ≥ 35 cm.	Low risk

Footnote: the exposure of the included studies assessed by maternal recall.

**Table 2 ijerph-21-00856-t002:** Findings of included studies conducted in the Africa region from 1994 to 2023.

Adverse Birth Outcome						
No	Author (Year)	Exposure	Outcome	Statistical Method	Effect Estimates (OR, RR, HR) (95% CI)	
	Zelka 2023 [58]	IFA supplementation	Adverse pregnancy outcomes	Binary logistic regression	No	Ref
					IFA	aOR 0.52 (0.41 to 0.68)
	Hailemichael 2020 [59]	IFA supplementation	Adverse birth outcome	Binary logistic regression	No	Ref
					IFA	aOR 0.67 (0.18 to 2.57)
	Kone 2018 [63]	IFA supplementation	Fatal foetal outcomes	Binary logistic regression	IFA	Ref
					No	aOR 3.15 (1.71 to 5.80)
Stillbirths						
	Chalumeau 2002 [66]	Iron supplementation	Stillbirths	Binary logistic regression	No	1
					Iron	Not significant on crude association
	Lolanso 2021 [60]	IFA supplementation	Stillbirths	Binary logistic regression	No	aOR 8.26 (4.82 to 14.16)
					IFA	1
	Ibrahim 1994 [62]	Iron supplementation	Stillbirths	Binary logistic regression	No	Ref
					Iron	No significant effect (not reported)
	Hailemichael 2020 [59]	IFA supplementation	Adverse birth outcome (stillbirths)	Binary logistic regression	No	Ref
					IFA	aOR 0.67 (0.18 to 2.57)
	Kone 2018 [63]	IFA supplementation	Fatal foetal outcomes (stillbirths)	Binary logistic regression	IFA	Ref
Perinatal death						
	Goba 2017 [33]	Iron supplementation	Perinatal mortality	Binary logistic regression	Iron	Ref
					No	aOR 3.30 (1.16 to 9.76)
	Ali 2014 [61]	Iron supplementation	Perinatal mortality	Binary logistic regression	No	Ref
					Iron	aOR 0.06 (0.02 to 0.10)
	Ibrahim 1994 [62]	Iron supplementation	Perinatal death	Binary logistic regression	No	Ref
					Iron	No significant effect (not reported)
	Kone 2018 [63]	IFA supplementation	Fatal foetal outcomes (perinatal)	Binary logistic regression	IFA	Ref
Neonatal death						
	Titaley [65]	IFA supplementation	Neonatal mortality	Cox proportional hazard ratio	No	Ref
					IFA	aHR 0.81 (0.71 to 0.94)
	Godha 2022 [64]	IFA supplementation	Neonatal mortality	Binary logistic regression	No	Ref
					IFA	Negative association in bivariable association but lost in the final multivariate model
	Ibrahim 1994 [62]	Iron supplementation	Neonatal death	Binary logistic regression	No	Ref
					Iron	No significant effect (not reported)

Footnote: IFA: Iron Folic Acid, aOR: Adjusted Odds ratio, aHR: Adjusted Hazard Ratio.

## Data Availability

Not applicable.

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
