# Peer review of "The Effects of Maternal Iron and Folate Supplementation on Pregnancy and Infant Outcomes in Africa: A Systematic Review"

_ijerph, 2024, doi:10.3390/ijerph21070856_

Round 1
Reviewer 1 Report
Comments and Suggestions for Authors
Considering SES and health disparities, this systematic review focused on iron-only, folate-only, and IF supplementation for perinatal health in Africa. The manuscript addresses an important public health topic for the African continent and other middle—and low-income countries. The manuscript needs to be reviewed for punctuation and could benefit from a review to improve the flow of the text, for example, by combining phrases into single paragraphs.
Even though the authors focused on supplementation only and justified the reason for excluding fortification, it is important to add a paragraph to the discussion regarding the fortification of iron, especially folate, in countries of Africa. Is the absence of folate supplementation due to the food fortification programs in the analyzed countries? Additionally, it would enrich the manuscript if the authors could add a short paragraph about structural barriers to supplement use in the African continent; how much does the government invest in supplementation programs? Is there a cost for individuals to access supplements? Are they easy to get access to in the healthcare service?
Introduction:
Line 55-56: add the rate of mortality for Sub-Saharan Africa.
Lines 58-68: Change the font size to match the rest of the text.
Line 63: add per 1000 live births.
Line 70: add references.
Methods:
Lines 156-158: Please combine the individual phrases into a single, coherent paragraph to improve the flow of the section and make it easier for the reader to follow your methodology.
Lines 160 – 166: Combine the phrases in a single paragraph.
Lines 173 -179: Combine the phrases in a single paragraph.
Lines 181-182: Combine the phrases in a single paragraph.
Lines 184-191: What was the time range for literature search? Past ten years? Five years? Be specific.
Results:
Figure 1: adjust the font size and image. The arrows are not aligned.
Lines 236-238: List which multiple countries from Sub-Saharan and West Africa were included.
Line 238: Missing an end-of-phrase mark after the word “Africa.”
Line 240: remove the end of the phrase mark after Table 1 inside the parenthesis.
Lined 241-244: How were rural and urban defined in these studies? Were the same parameters used?
Lines 293-298: I know the authors' interest was in supplementation only, but I wonder if the lack of folate supplementation is due to the fortification of food with folate. This would be a good topic to include in the discussion.
Lines 330-332: The phrase needs to be completed. Please revise.
Line 380: Missing number for the subheading: perinatal death
Lines 396-397: The authors have referred to information on high-malaria endemic areas for the first time. Why is this information necessary for the explored outcome? If it is not relevant, remove it.
Discussion:
Lines 424-431: I suggest to the authors that they consider naming the analysis a scoping review. This is an adequate methodology for themes that do not have as much literature as the one explored here yet inform the direction of needed research.
Lines 443-446: add references. Also, reframing should be considered as it did not consider the effect of social determinants of health.
Lines 468-470: add references.
Comments on the Quality of English LanguageThe manuscript needs to be reviewed for punctuation and could benefit from a review to improve the flow of the text, for example, by combining phrases into single paragraphs.
Author Response
|
Summary |
|||
|
We sincerely thank the reviewers for their thorough review of the manuscript and for providing constructive feedback. We believe we have thoroughly incorporated their comments into the updated version. |
|||
|
Point-by-point response |
|||
|
Page numbers |
Line numbers |
Reviewers Comment |
Revised versions |
|
2 |
57 to 60 |
Line 55-56: add the rate of mortality for Sub-Saharan Africa.
|
Thank you for your valuable suggestion. We have included the rate of morality in the updated manuscript.
“However, the reduction in neonatal mortality has been much slower in Africa, particularly in the Sub-Saharan region, decreasing from 45.3 per 1000 live births in 1990 to 27.5 per 1000 live births in 2019” |
|
2 |
64 to 75 |
Lines 58-68: Change the font size to match the rest of the text.
|
Based on your recommendation, we have corrected the font size in the updated manuscript.
“In Africa, adverse birth outcomes, neonatal mortality, and infant mortality remain a main challenge, with variations across regions. For example, in Nigeria, the neonatal mortality rate was stagnant at 41 per 1000 live births from 1990 to 2013, with rural areas facing higher rates (42 per 1000 live births) than urban areas (31 per 1000) (4). Similarly, in Ghana, rural areas had significantly higher neonatal deaths at 63.6 per 100 compared to 36.4 per 100 in urban areas (5). Another study in Nigeria found infant mortality rates of 70 per 1000 live births in rural areas and 49 per 1000 live births in urban areas (6). This persistent challenge hinders the region’s progress toward national and international goals, particularly in rural areas where access to healthcare and enablers of health literacy are much more challenging.”
|
|
2 |
68 to 70 |
Line 63: add per 1000 live births.
|
Thank you for your careful review and for identifying this important point. We have rectified this in the updated manuscript.
“Similarly, in Ghana, rural areas had significantly higher neonatal deaths at 63.6 per 1000 live births compared to 36.4 per 1000 live births in urban areas (5). ” |
|
2 |
75 |
Line 70: add references. |
Thank you for your suggestion. We have now added references to the statement in the updated manuscript.
“This persistent challenge hinders the region’s progress toward national and international goals, particularly in rural areas where access to healthcare and enablers of health literacy are much more challenging (7, 8).” |
|
4 |
183 to 188 |
Lines 156-158: Please combine the individual phrases into a single, coherent paragraph to improve the flow of the section and make it easier for the reader to follow your methodology. |
Thank you for your insightful suggestion to combine these sections. We have implemented this change in the revised manuscript.
“This review included studies that investigated the effects of oral iron-only, folate-only, and IFA supplementation during pregnancy. Studies involving fortification, combination with other micronutrients, and measured serum iron or folate levels as the intervention or exposure were excluded from the review as we were primarily interested in oral exposure's effects, given its universal comparison across different countries.” |
|
4 |
190 to 191 |
Lines 160 – 166: Combine the phrases in a single paragraph. |
We merged the phrases in the updated manuscript. “The comparator group consisted of women who either took a placebo or did not receive any micronutrient supplementation.” |
|
4 and 5 |
194 to 202 |
Lines 173 -179: Combine the phrases in a single paragraph. |
Based on your suggestion, we combined the phrases in the updated manuscript as follows:
“Adverse birth outcomes encompass a range of complications that can occur during pregnancy or childbirth. These include stillbirth (pregnancy termination after 28 weeks), small for gestational age (birth weight below the 10th percentile for gestational age), preterm birth (delivery before 37 weeks), low birth weight (birth weight less than 2500 grams), perinatal mortality (fetal death after 28 weeks or newborn death within the first week), and neonatal mortality (newborn death within the first 28 days) (52, 53). Notably, adverse birth outcomes are assessed using a combination of more than two of these indicators.”
|
|
5 |
204 to 205 |
Lines 181-182: Combine the phrases in a single paragraph. |
Thank you for your suggestion. We have combined the phrases in the updated manuscript, as you recommended:
“Neonatal death is defined as mortality within the first 28 days, while infant mortality occurs within the first year (52, 53).” |
|
5 |
207 to 215 |
Lines 184-191: What was the time range for literature search? Past ten years? Five years? Be specific. |
To address the limited evidence available in the region, we did not limit our search with time in the electronic databases. We have also added a relevant statement to the revised manuscript:
“The relevant studies were retrieved from the following electronic databases on 29/08/2023: MEDLINE, PsycINFO, Embase, Scopus, CINAHL, Web of Science, and Cochrane. No time restrictions were applied during the search in these databases. Additionally, Google Scholar and Google Advanced Search were utilised to access grey literature and additional literature. The final search term used for the MEDLINE database to retrieve relevant articles is attached as a supplementary file (Table S2). The last Google Scholar search was conducted on 07/03/2024 to ensure that the studies included were as recent as possible.” |
|
7 |
283 to 322
|
Figure 1: adjust the font size and image. The arrows are not aligned. |
Based on your suggestion, we corrected the font size and arrow alignment.
“Figure 1. PRISMA (Preferred Reporting Items for Systematic Review and Meta-Analysis) flow chart diagram of studies searched for systematic review and meta-analysis in Africa, 2023.” |
|
6 |
258 to 266 |
Lines 236-238: List which multiple countries from Sub-Saharan and West Africa were included. |
We have included the lists of countries in the manuscript, as you recommended:
“A total of 10 studies were eligible for inclusion. Of which, four were conducted in Ethiopia (33, 58-60), two in Sudan (61,62), one in Cote d’Ivoire (60) and three involved multiple countries in the sub-Saharan Africa region (Zimbabwe, Malawi, Cameroon, Uganda, Burkina Faso, Madagascar, Tanzania, Liberia, Benin, Congo, Niger, Zambia, Senegal, Chad, Guinea, the Democratic Republic of the Congo, Ghana, Mali, Nigeria, Angola, Burundi, Gambia, Kenya, Mauritania, and Sierra Leone) and the West Africa region (Ivory Coast, Mali, Senegal, Niger, Mauritania and Burkina Fano) (64-66).” |
|
6 |
265 |
Line 238: Missing an end-of-phrase mark after the word “Africa.” |
Thank you for your valuable input. We have revised this statement accordingly (Please refer to above). |
|
6 |
268 |
Line 240: remove the end of the phrase mark after Table 1 inside the parenthesis. |
Based on your suggestion, we removed it in the revised manuscript.
“Regarding study design, four were cross-sectional (60,61,64,65), two were case-control (33,59), and four were cohort studies (58, 62, 63, 66) (Table 1).” |
|
6 |
272 to 276 |
Lined 241-244: How were rural and urban defined in these studies? Were the same parameters used? |
Thank you for your suggestion. However, none of the included articles defined urban/rural. For this, we included the following information in the updated manuscript.
“Even though studies did not include urban-rural definitions, the criteria commonly employed in the region include lifestyle, population density, economic activities, infrastructures, settlement patterns, industrialisation, and correlation with poverty (67).” |
|
21 |
523 to 530 |
Lines 293-298: I know the authors' interest was in supplementation only, but I wonder if the lack of folate supplementation is due to the fortification of food with folate. This would be a good topic to include in the discussion. |
Thank you for your input. As you suggested, we have incorporated information about folate in the discussion section of the revised manuscript.
“In general, there is a lack of studies conducted in Africa, and the study design employed often falls within the lower level of the evidence hierarchy. Additionally, no evidence was found regarding the impact of iron-only, folate-only, and IFA supplementation on small for gestational age and infant mortality. This lack of evidence may impede the data generation decisive for informing policy decisions. Interestingly, the absence of evidence on the impact of folic acid supplementation itself may indicate folic acid in this region being primarily available through fortified food sources, warranting further investigation. |
|
21 |
531 to 539 |
Additionally, it would enrich the manuscript if the authors could add a short paragraph about structural barriers to supplement use in the African continent; how much does the government invest in supplementation programs? Is there a cost for individuals to access supplements? Are they easy to get access to in the healthcare service? |
Thank you for your suggestion. Based on your recommendation, we have added a paragraph in the updated manuscript:
“Some African countries offer free iron and/or folate supplementation programs (69). However, program effectiveness is hampered by limited access. This includes the insufficient supply of supplements, inadequate counselling on their benefits and appropriate use, and challenges for pregnant women to adhere to supplementation schedules (70). Furthermore, structural barriers also hinder uptake. Lack of leadership engagement, weak family support systems, and limited access to antenatal care services restrict program reach. Additionally, misconceptions about supplementation, fear of side effects, and social norms that discourage its use create further hurdles that should be addressed as there is a clear need for improvements in health literacy (71, 72).” |
|
13 |
362 to 366 |
Lines 330-332: The phrase needs to be completed. Please revise. |
Thank you for your suggestion. We have revised the entire sentence.
“Two of the three studies found positive associations between oral IFA supplementation and reduced adverse birth outcomes. One study reported an adjusted odds ratio (aOR) of 0.52 (95% CI: 0.41 to 0.68) (58). The other study found an aOR of 3.15 (95% CI: 1.71 to 5.80) (63), indicating a possible protective effect of IFA supplementation.” |
|
14 |
414 |
Line 380: Missing number for the subheading: perinatal death |
Thank you for pointing this out. We have added the missing information in the updated manuscript.
“3.4.1.3. Perinatal death (n=4)” |
|
14 |
430 to 433 |
Lines 396-397: The authors have referred to information on high-malaria endemic areas for the first time. Why is this information necessary for the explored outcome? If it is not relevant, remove it. |
Thank you for your suggestion. Based on it, we have removed the “high malaria endemic area” from the updated manuscript.
“Two out of three studies (64, 65) conducted in sub-Saharan countries employed data from large, nationally representative Demographic and Health Surveys (DHS) to examine the relationship between oral IFA supplementation and neonatal mortality. ” |
|
|
|
Lines 424-431: I suggest to the authors that they consider naming the analysis a scoping review. This is an adequate methodology for themes that do not have as much literature as the one explored here yet inform the direction of needed research. |
Thank you for suggesting the terminology to describe our analysis (lines 424-431). We appreciate you highlighting the similarities between our approach and a scoping review, particularly in terms of mapping the available literature.
While acknowledging the similarities, we opted to maintain the systematic review classification because our methodology incorporated additional elements commonly associated with systematic reviews. These elements strengthen the rigor and transparency of our analysis. We developed a pre-defined protocol outlining the research question, inclusion/exclusion criteria, and search strategy before commencing the review (a priori protocol). Furthermore, we employed a comprehensive search strategy across multiple electronic databases and other relevant sources to identify all potential studies. Finally, we independently assessed the methodological quality of the studies included in the review using established criteria using Cochrane Effective Practice and Organisational Care (EPOC) [55] and the Newcastle-Ottawa Scale (NOS) [56] risk of bias tool. These elements, alongside the mapping of the available literature, contribute to a robust systematic review of the research on this topic. |
|
20 |
477 to 480 |
Lines 443-446: add references. Also, reframing should be considered as it did not consider the effect of social determinants of health. |
Thank you for your suggestion. We have revised the sentence to address the limitation of lacking stratification analysis by social determinants of health, as you pointed out:
“None of the included studies performed a stratification analysis by variables representing social determinants of health (58,59,63). This is important because factors like urban/rural residence, educational status, age, and income can potentially influence the effects of oral IFA supplementation during pregnancy on birth outcomes.” |
|
19 |
504 |
Lines 468-470: add references. |
Thank you for your comment regarding referencing. We have incorporated references in the revised manuscript.
“This urban-rural participant difference between the included studies may lead to differences in service accessibility and health literacy (33,61).” |
|
|
|
The manuscript needs to be reviewed for punctuation and could benefit from a review to improve the flow of the text, for example, by combining phrases into single paragraphs. |
Thank you for your valuable suggestion. The senior authors have now fixed the grammatical errors. |

Reviewer 2 Report
Comments and Suggestions for Authors
1. The article “The effects of maternal oral iron and folate supplementation on pregnancy and infant outcomes in Africa: A systematic review” intend to assess individual and combined effects of iron and folate intake during pregnancy on infant outcomes in Africa. Although global data are available on this issue, the impact of IFA supplementation can vary with geography and ethnicity and, therefore, could be interesting if presented rationally. Unfortunately, the article in its present form is not suitable for publication for several reasons. A few of these comments are listed below to revise the manuscript carefully and logically.
2. The rationale and purpose of the article must be mentioned in the abstract, mainly why each of these micronutrients was chosen individually. If the individual impact of this micronutrient is described, it should be compared with a placebo.
3. In lines 27-28, the statement “Our qualitative synthesis analyzed ten articles reporting….” does not clearly state to whom adverse effects were pointed out.
4. Line 31-34 conclusion- each statement contradicted by the next statement. For example.
First, “iron-only supplementation during pregnancy could reduce perinatal death in African women’’. Again, “Limited evidence exists for iron-only and IFA supplementation in reducing……’’
5. Introduction- should add data on IFA and its impact on infant outcome from global data, particularly with closer ethnicity and geography, wherever possible.
6. Fig.1 The qualitative attribute like “wrong statement” or “wrong comparator” should be defined.
7. Table 1 should include a column with the objective for each study. Remove column exposure assessment, and include this information in the footnote. Merge country and year.
8. The title should omit the ‘oral’ term.
9. The study is biased in its focus on the outcome rather than the causality of the hypothesis, as also reflected in the reference section, which places a major emphasis on adverse pregnancy outcomes (references 1 to 24).
10. The study objective should narrate global literature/ WHO recommendation (https://www.who.int/data/nutrition/nlis/info/antenatal-iron-supplementation) about the risk or benefit or their ratio of each micronutrient on adverse pregnancy outcome and infant outcome.
11. In several places in the article, the authors recommend iron-only supplementation during pregnancy. However, the women’s iron and haemoglobin levels (anaemia status) must be known in such cases. Without adding such attributes, these statements would be misleading for the stakeholders to follow.
12. Moreover, according to WHO, folic acid supplementation (with or without iron) provided before conception and during the first trimester of pregnancy is also recommended to decrease the risk of neural tube defects.
13. The authors must follow a more neutral tone in suggesting and recommending from the limited data accessible to them.
Comments on the Quality of English Language
see before
